# Prevalence and evolutionary analyses of human T-cell lymphotropic virus in Guangdong province, China: Transcontinental and Japanese subtype lineages dominate the prevalence

Qiao Liao[1,2]☯, Zhengang Shan[1,2]☯, Min Wang[1,2], Jieting Huang[1,2], Ru Xu[1,2], Tingting Li[3], Wenjing Wang[3], Chengyao Li[3], Xia Rong[1,2,3]*, Yongshui Fu[1,2,3,4]*

**1** Guangzhou Blood Center, Guangdong, China, **2** The Key Medical Disciplines and Specialties Program of Guangzhou, Guangdong, China, **3** School of Laboratory Medicine and Biotechnology, Southern Medical University, Guangzhou, Guangdong, China, **4** Zhujiang Hospital of Southern Medical University, Guangzhou, Guangdong, China

☯ These authors contributed equally to this work.
* joyjoy@126.com (XR); fuyongshui1969@yahoo.com (YF)

**Data Availability Statement:** All relevant data are within the manuscript and its Supporting Information files.

## Abstract

To systematically characterize the prevalence and evolution of human T-cell lymphotropic virus (HTLV) infection among voluntary blood donors (BDs) in Guangdong province, China. A three-year survey for HTLV epidemiology among BDs was performed in Guangdong during 2016–2018. Anti-HTLV-1/2 was screened by ELISA and ECLIA, and subsequently confirmed by western blot (WB) and nucleic acid testing (NAT). The prevalence of HTLV in donors from different cities was calculated. The identified HTLV-positive cases were phylogenetically genotyped and analyzed in a Bayesian phylogenetic framework. Among 3,262,271 BDs, 59 were confirmed positive for HTLV-1 (1.81 per 100,000) and no HTLV-2 infection was found. The prevalence of HTLV-1 varied significantly among 21 cities in Guangdong province, China. The highest prevalence was found in donors from Shanwei (13.94 per 100,000), which is a coastal city in eastern Guangdong. Viral genomic sequences genotyped from 55 HTLV-1 carriers showed that 39 were transcontinental subtype and 16 were Japanese subtype. Specially, 13 out of 39 transcontinental subtype sequences were characterized with L55P mutation and 21 out of 55 sequences were characterized with L19F mutation in viral gp46 protein. The L55P mutation seemed be specific to eastern Asia since it only presented in the sequences from Japan, mainland China, and Taiwan. Phylogenetic analysis of gp46 gene shows that HTLV-1a may have been introduced to Guangdong through four different introduction events and formed major transmission clusters: clades I (13,602 years ago), II(16, 010 years ago), III(15,639 years ago) and IV(16,517 years ago). In general, Guangdong is considered to be a low-prevalence region for HTLV-1 infection, but the prevalence is significantly higher in Shanwei city. Transcontinental and Japanese subtype lineages dominate the prevalence in Guangdong. In terms of blood safety, HTLV

**Funding:** The authors have received funding from the following sources: the Weigao Fund of the Chinese Society of Blood Transfusion (XR), CSBT-WG-2018-08, National Science and Technology major projects (YF), 2018ZX10302205-001, Guangdong Medical Science and Technology Research Fund Project (QL), A2018466, the National Natural Science Foundation of China (YF), 81772208, the Key Medical Laboratory of Guangzhou from the Key Medical Disciplines and Specialties Program of Guangzhou, Guangdong, China. The funders had no role in study design, data collection and analysis, decision to publish, or preparation of the manuscript.

**Competing interests:** The authors have declared that no competing interests exist.

antibody screening for first-time blood donors can effectively reduce the risk of HTLV transmission.

## Author summary

Human T-cell lymphotropic virus type 1 distributed all over the world. Since 1988, serological screening has been included in routine blood screening in certain developed countries and regions such as American countries and some parts of Western Europe and East Asia. However, data from some highly populated countries such as China are still not available. We performed a 3-year large-scale blood screening survey to systematically characterize the prevalence of HTLV infection among blood donors in Guangdong province in south China during 2016–2018. In general, Guangdong was considered to be a low-prevalence region for HTLV-1 infection, but the prevalence is significantly higher in Shanwei, a coastal city of eastern Guangdong. Transcontinental and Japanese subtype lineages dominate the prevalence in Guangdong. Moreover, similar molecular characteristics of prevalent HTLV-1 sequences in Mainland China, Taiwan and Japan suggested a same origin of these viruses.

## Introduction

The World Health Organization (WHO) published an open letter in May 2018 calling for global efforts to eradicate human T-cell lymphotropic virus type 1(HTLV-1) infection [1], which causes cancer and several other diseases such as adult T-cell leukemia/lymphoma and HTLV-1 associated myelopathy/tropical spastic parparesis. Globally, the HTLV-1 infected population is up to 10 to 15 million [2]. Similar to human immunodeficiency virus (HIV), HTLV can be transmitted through blood, sexual contact, and breast feeding [3].

HTLV-1 is mainly prevalent in the Caribbean, South and Central America, Central and Western Africa, Southwestern Japan, Australia and Melanesia[4]. HTLV-2 has a more restricted distribution than HTLV-1, and is predominately prevalence among some Native Americans and Central African tribes. HTLV-2 is relatively common among intravenous drug users and their sex partners in Europe, North America, and other regions of the world [5]. The majority of HTLV-infected individuals are asymptomatic carriers, so the risk of transfusion transmission should be considered. Since 1988, serological screening has been included in routine blood screening in certain developed countries and regions such as American countries and some parts of Western Europe and East Asia [6]. The WHO has recommended that the determination concerning the screening of HTLV in BDs should be guided by local epidemiological evidence. However, except for Iran and Japan, epidemiological data regarding HTLV infection seems to be rare in most Asian countries. Moreover, large-scale and representative studies, especially in populated countries like China and India, are critical for the eradication of the virus. Herein, we aimed to investigate the prevalence of HTLV among BDs in Guangdong province, which has the largest resident population and immigrant population of more than 100 million in China.

## Materials and methods

### Ethic statement

During 2016–2018, a total of 3,262,271 blood samples of BDs were collected from 24 blood stations in 21 cities of Guangdong province. The participants have provided written informed

consent before enrollment. The study protocol was according to the ethical guidelines of the 1975 Declaration of Helsinki and was approval by Guangzhou Blood Center Medical Ethics Committee, Grant No.006, 2016.

### Serological testing

Blood samples were screened for HTLV-1/2 infection using an enzyme-linked immunosorbent assay (ELISA) kit from one of the three following manufacturers: (1) Beijing Wantai Biological Pharmacy Enterprise Co., Beijing, China; (2)DiaSorin S.p.A UK Branch; or (3)Abbott Laboratories, USA. The assays had been previously validated to be sensitive and specific [7]. All reactive samples in the initial screening were subjected to a repeated test. When a repeated test was also reactive, the samples were subsequently verified using an electrochemiluminescence immunoassay (ECLIA: Roche, German). ECLIA-positive samples were further tested by western blot (HTLV BLOT 2.4, MP Biomedicals SAS, U.S) to confirm the presence of anti-HTLV-1 and/or HTLV-2 antibody.

### DNA extraction and nucleic acid testing

Proviral DNA of anti-HTLV positive samples were extracted from peripheral blood mononuclear cells using the QuickGene-610L system (Fujifilm, Tokyo, Japan) following the manufacturer's instructions. The HTLV nucleic acid was detected using a real-time PCR assay (Fluorescence probing-PCR, Beijing Wantai Biological Pharmacy Enterprise Co., Beijing, China) following manufacturer's instructions. The real-time PCR positive samples were amplified for the coding region of the HTLV-1 gp46 gene and LTR gene by nested-PCR[8]. The primers for the gp46 region (nt5222-nt6151) were as follows: outer primers F1(nt 5031–5051), 5'-AGC CGC CAG TGG AAA GGA CCA-3'; R1(nt6344-6321), 5'- CCT CGT CTG TTC TGG GCA GCA TAC-3'; inner primers F2 (nt5203-5222), 5'—ATGGGTAAGT TTCTCGCCAC -3'; R2 (nt6168-6151), 5' -GGA GAC AAG CCA GGC CGC -3'. The first-round PCR consisted of initial denaturation at 95˚C for 10 min; 25 cycles of 95˚C for 50 s, 52˚C for 50 s, 72˚C for 90 s; 72˚C for 10min. The second-round PCR consisted of 95˚C for 5 min; 25 cycles of 95˚C for 50 s, 55˚C for 50 s, and 72˚C for 90 s; 72˚C for 10min. The primers for the LTR region (nt8271-nt9043) were as follows: outer primers F1(nt8168-8188), 5'-CCC TCA TTT CTA CTC TCA CAC-3'; R1(nt9022-9043), 5'-TGT GTA CTA AGT TTC TCT CCT G-3'; inner primers F2(nt8271-8292), 5'-ACG AAA AAG AGG CAG ATG ACA A-3'; R1(nt9022-9043), 5'-TGT GTA CTA AGT TTC TCT CCT G-3'; The first-round PCR consisted of initial denaturation at 95˚C for 5 min; 30 cycles of 95˚C for 60 s, 55˚C for 60 s, 72˚C for 60 s; 72˚C for 10min. The second-round PCR consisted of 95˚C for 5 min; 30 cycles of 95˚C for 60 s, 59˚C for 60 s, and 72˚C for 60 s; 72˚C for 10min.The PCR products were purified, either directly or by clone sequencing (The Beijing Genomics Institute, Shenzhen, China).

### Genotype determination and phylogenetic analysis

The resulting sequences were aligned using BioEdit 5.0.9 (http://www.mbio.ncsu.edu/BioEdit/bioedit.htlm). Phylogenetic trees were constructed by the maximum-likelihood (ML) method in MEGA7.0. The ML corrected distances (using the HKY+I+Γ6 substitution mode) were determined with 1000 iterations of bootstrap sampling. HTLV genotype reference sequences were retrieved from GenBank. The new nucleotide sequence data reported in this paper were deposited in the GenBank databases under the accession numbers MT113123- MT113153 and MT113155-MT113178.The Bayesian Markov chain Monte Carlo (MCMC) inference method, implemented in BEAST v1.10.4, was used to analyze the origin, evolution, and transmission of HTLV-1 in Guangdong BDs, using gp46 and LTR sequences from this study and existing

reference sequences from other countries. We found that the time-dependent rate model with Markov jumps process and yule.birthRate clock model was the best-fitting comparing with the strict and uncorrelated lognormal models based on the Bayes factors[9,10]. The independent GTR+G+I model was used for each codon, which allowed us to estimate relative rates of evolution at each codon position. These analyses employed independent GTR+G+I substitution models, LTR gene for the 2nd codon positions, and gp46 gene for the 3rd codon positions. The time of the most recent common ancestor (tMRCA) for the HTLV-1 subtypes was estimated also using the time-dependent rate model and the yule.birthRate clock. We used the following prior rates, as described previously[11]:$2.1\times10^{-7}$ ($2.1\times10^{-8}$ to $4.5\times10^{-7}$) substitutions/site/year and $5.6\times10^{-7}$ ($2.1\times10^{-8}$ to $4.5\times10^{-7}$) substitutions/site/year for gp46 and LTR, respectively. We managed sufficient chain length for all MCMC processes that finally generated 10,000 trees for HTLV subtype. Using the Tracer program, the sampling convergence was assessed by the estimated effective sampling size (ESS). In this study, when all ESS numbers were $\geq$200 sufficient sampling was considered to have been achieved according to the manual of BEAST software.

## Statistical analysis

All statistical analyses were performed using SPSS 16.0 software (SPSS, Chicago, IL, USA). For donors with multiple donations during the period, only the screening results of the first sample were calculated. Statistical analyses were performed by the Mantel-Haenszel $\chi$2 test and Fisher's exact test for categorical variables. A P-value of <0.05 was considered to be statistically significant.

## Results

### Screening of HTLV from blood donors in Guangdong province

A total of 3,262,271 BD samples were tested for anti-HTLV by ELISA from March 2016 to December 2018 in 24 blood stations in Guangdong province, China. It was found that 1,122 samples were reactive by ELISA. Among them, 59 were confirmed positive for anti-HTLV-1, by WB and NAT (Fig 1). No anti-HTLV-2 positive samples were detected.

The 59 HTLV-1 carriers came from 16 cities in Guangdong province. The overall infection rate of HTLV-1 was 1.81 per 100,000 (59/3, 262,271). We found that eastern Guangdong had the highest prevalence of HTLV-1 (5.33 per 100,000), followed by western Guangdong (2.54 per 100,000). The HTLV-1 prevalence rate in eastern Guangdong was significantly higher than that in the Pearl River Delta area (P<0.05). We also found that the prevalence in Shanwei, a coastal city in eastern Guangdong, was the highest (13.94 per 100,000), followed by Zhanjiang (5.31 per 100,000), a coastal city in southwestern Guangdong. The detailed information on cities and HTLV-1 prevalence is provided in Table 1, some of these data refer to our earlier studies[12].

### Demographic characteristics of HTLV-1 infected blood donors

The demographic information (age, sex, donation history and ethnicity) of 59 confirmed HTLV-1 infected BDs is shown in Table 2. The infected donors were born in eight different provinces, of which five were non-coastal areas. However, all of the donors lived in southeastern coastal areas. Of the 46 donors born in Guangdong province, 19 were born in eastern Guangdong, 12 were born in western Guangdong, 14 were born in Pearl River Delta area and only 1 was born in northern Guangdong. Of the additional 13 donors who were not born in Guangdong, 7 were born in Fujian province, 1 was born in Zhejiang province, 3 were born elsewhere in southwestern China, and 2 were born in central China.

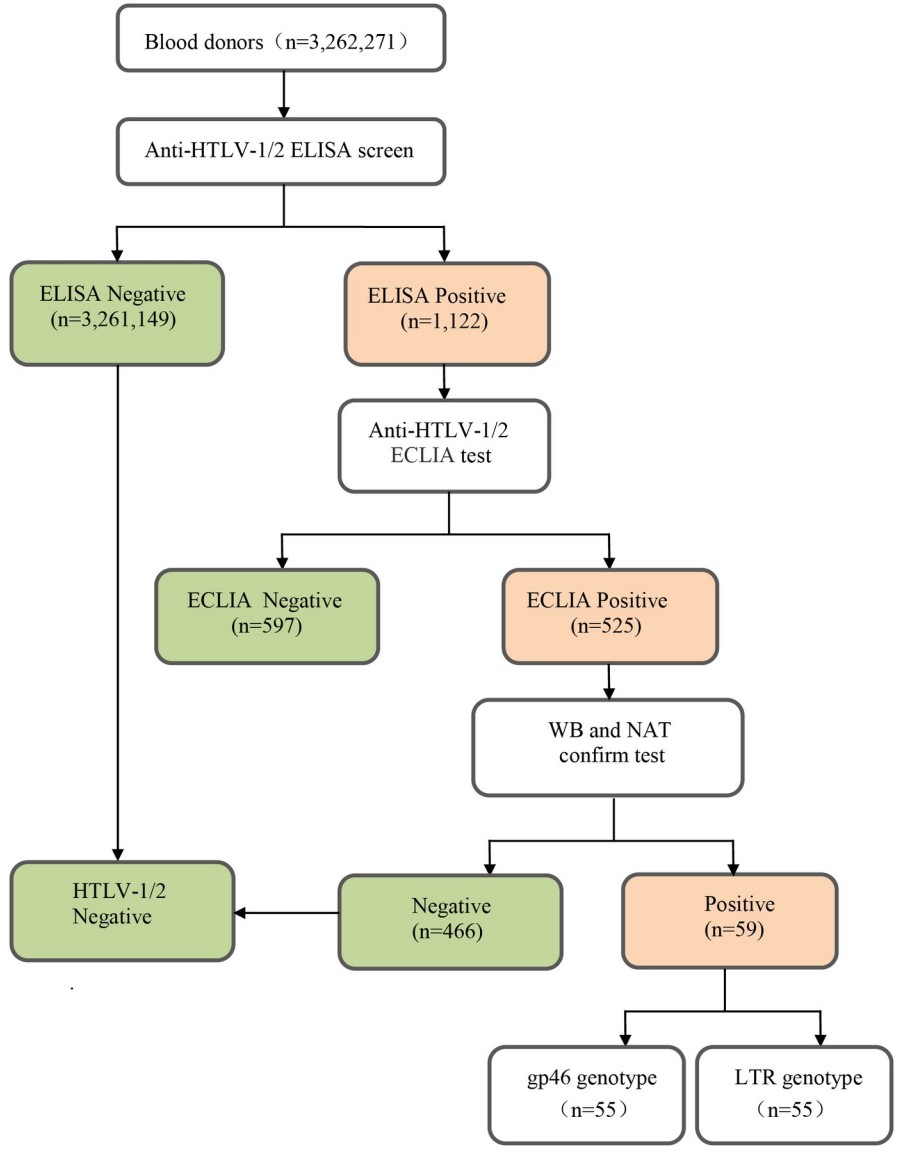

**Fig 1. Flow chart of screening and confirmation of HTLV infection.**

## Phylogenetic analysis of HTLV-1 prevalence in Guangdong province

The HTLV-1 gene was highly conserved. The homology of different isolates ranged from 91% to 100%, mainly including four genotypes of HTLV-1: HTLV-1a (Cosmopolitan), 1b (Central African), 1c (Melanesian), and 1d (Pygmies); HTLV-1a also contained three main subgroups: Transcontinental, Japanese, and West African. In this study, out of the 59 HTLV-1 confirmed positive specimens, 55 sequences of the gp46 and LTR genomic region were successfully obtained, which were used for genotyping by phylogenetic analysis. Genotyping of gp46 sequences showed 39 Transcontinental subtypes and 16 Japanese subtypes of HTLV-1a genotype (Fig 2), which was in accordance with the genotyping of LTR (S1 Fig).

To estimate the regions of origin and time of emergence of the HTLV-1a genotype in China, we performed phylogenetic analysis by constructing a maximum-clade-credibility

**Table 1. Demographic characteristics of the donors in Guangdong.**

| Geographic regions | Sample, NO. | Confirmed HTLV (+) | | OR(95%Cl) | P value |
|---|---|---|---|---|---|
| | | Sample, NO. | Prevalence (per 100,000) | | |
| **Place** | | | | | |
| **Overall** | 3,262,271 | 59 | 1.81 | 1 | |
| **Pearl River Delta** | | | | | |
| Guangzhou | 1,014,424 | 11 | 1.08 | 0.6(0.32–1.14) | 0.071 |
| Foshan | 326,122 | 6 | 1.84 | 1.02(0.44–2.36) | 0.546 |
| Dongguan | 170,672 | 3 | 1.76 | 0.97(0.31–3.1) | 0.628 |
| Huizhou | 191,181 | 5 | 2.62 | 1.45(0.58–3.6) | 0.28 |
| ZhuHai | 59,378 | 1 | 1.68 | 0.93(0.13–6.72) | 0.709 |
| Shenzhen | 139,147 | 4 | 2.87 | 1.59(0.58–4.38) | 0.257 |
| Jiangmen | 118,185 | 2 | 1.69 | 0.94(0.23–3.83) | 0.64 |
| Zhongshan | 137,048 | 1 | 0.73 | 0.4(0.06–2.91) | 0.298 |
| Zhaoqing | 110,982 | 0 | 0 | 0 | 0.139 |
| **East Guangdong** | | | | | |
| Chaozhou | 24,645 | 0 | 0 | 0 | 0.641 |
| Shantou | 85,557 | 3 | 3.51 | 1.94(0.61–6.19) | 0.211 |
| Shanwei | 43,027 | 6 | 13.94 | 7.71(3.33–17.86) | <**0.001**$^*$ |
| Jieyang | 90,650 | 4 | 4.41 | 2.44(0.89–6.72) | 0.091 |
| **West Guangdong** | | | | | |
| Yunfu | 70,134 | 1 | 1.43 | 0.79(0.11–5.69) | 0.639 |
| Yangjiang | 64,965 | 0 | 0 | 0 | 0.312 |
| Maoming | 147,859 | 2 | 1.35 | 0.75(0.18–3.06) | 0.504 |
| Zhanjiang | 150,629 | 8 | 5.31 | 2.94(1.4–6.15) | **0.009**$^*$ |
| **North Guangdong** | | | | | |
| Shaoguan | 154,946 | 1 | 0.65 | 0.36(0.05–2.58) | 0.238 |
| Heyuan | 42,440 | 0 | 0 | 0 | 0.466 |
| Meizhou | 94,374 | 1 | 1.06 | 0.59(0.08–4.23) | 0.494 |
| Qingyuan | 25906 | 0 | 0 | 0 | 0.627 |
| **Regions** | | | | | |
| Pearl River Delta | 2,267,139 | 33 | 1.46 | 1 | |
| East Guangdong | 243,879 | 13 | 5.33 | 3.66(1.93–6.96) | <**0.001**$^*$ |
| West Guangdong | 433,587 | 11 | 2.54 | 1.74(0.88–3.45) | 0.084 |
| North Guangdong | 317,666 | 2 | 0.63 | 0.43(0.1–1.8) | 0.179 |

(MCC) tree using the gp46 (Fig 3) and LTR (S2 Fig) sequences. The structures of gp46 and LTR gene MCC trees were similar. The tMRCA of the gp46 gene was 18,202 years ago (95% CI: 4,913–68,681) for the HTLV-1a genotype in China. Phylogeographic tree analysis of HTLV-1a genotype revealed different relatedness of the isolates from China and those from other countries. Based on the structure of the tree, the HTLV-1a sequences can be roughly divided into cluster A (Transcontinental subtype) and B (Japanese subtype). HTLV-1a may have entered Guangdong through four different introduction events, forming four major transmission clusters: clades I(13,602 years ago), II(16, 010 years ago), III(15,639 years ago) and IV(16,517 years ago). These findings suggest that HTLV-1a migrated to China and formed four major virus strains and some sporadic epidemic strains. Cluster B showed that the Japanese subtype cluster in Guangdong migrated from Japan.

In this study, two amino acid substitutions were found in the gp46 sequences obtained from the 55 HTLV-infected BDs in Guangdong. The characteristic L19F mutation in the signal

**Table 2.  Demographic characteristics of HTLV testing confirmed-positive donors.**

| Donor number | Sex | Age | Ethnicity | Birthplace | Residency | Number of blood donations |
|---|---|---|---|---|---|---|
| 1 | F | 47 | Han | Pearl River Delta | Guangzhou | 1 |
| 2 | M | 41 | Han | Hubei | Guangzhou | 1 |
| 3 | F | 22 | Han | Guizhou | Guangzhou | 1 |
| 4 | M | 31 | She | Fujian | Guangzhou | 1 |
| 5 | M | 20 | Han | East Guangdong | Guangzhou | 1 |
| 6 | F | 20 | Han | East Guangdong | Guangzhou | 1 |
| 7 | M | 23 | Han | East Guangdong | Guangzhou | 1 |
| 8 | F | 55 | Han | Pearl River Delta | Guangzhou | 1 |
| 9 | M | 55 | Han | Pearl River Delta | Guangzhou | 2 |
| 10 | M | 42 | Han | Pearl River Delta | Guangzhou | 2 |
| 11 | M | 39 | Han | Fujian | Guangzhou | 1 |
| 12 | F | 37 | Han | Pearl River Delta | Foshan | 5 |
| 13 | M | 47 | Han | Pearl River Delta | Foshan | 13 |
| 14 | F | 50 | Zhuang | Guizhou | Foshan | 1 |
| 15 | M | 35 | Han | West Guangdong | Foshan | 4 |
| 16 | M | 31 | Han | Pearl River Delta | Foshan | 1 |
| 17 | F | 34 | Han | Pearl River Delta | Foshan | 1 |
| 18 | M | 40 | Han | Fujian | Dongguan | 1 |
| 19 | F | 32 | Han | Fujian | Dongguan | 1 |
| 20 | F | 41 | Hui | Fujian | Dongguan | 1 |
| 21 | M | 53 | Han | East Guangdong | Huizhou | 2 |
| 22 | M | 23 | Han | East Guangdong | Huizhou | 2 |
| 23 | M | 53 | Han | Pearl River Delta | Huizhou | 1 |
| 24 | F | 35 | Han | East Guangdong | Huizhou | 1 |
| 25 | F | 43 | Han | Pearl River Delta | Huizhou | 1 |
| 26 | M | 35 | Han | ChongQing | Shenzhen | 1 |
| 27 | F | 30 | Yao | Hunan | Shenzhen | 2 |
| 28 | F | 60 | Han | Pearl River Delta | Zhuhai | 1 |
| 29 | M | 48 | She | Zhejiang | Shenzhen | 1 |
| 30 | M | 29 | Han | Fujian | Shenzhen | 1 |
| 31 | M | 42 | Han | Pearl River Delta | Jiangmen | 10 |
| 32 | F | 39 | Han | Pearl River Delta | Jiangmen | 1 |
| 33 | F | 48 | Han | Pearl River Delta | Zhongshan | 2 |
| 34 | M | 48 | Han | East Guangdong | Shantou | 1 |
| 35 | M | 44 | Han | Fujian | Shantou | 1 |
| 36 | M | 39 | Han | East Guangdong | Shantou | 1 |
| 37 | M | 30 | Han | East Guangdong | Shanwei | 2 |
| 38 | M | 35 | Han | East Guangdong | Shanwei | 1 |
| 39 | M | 43 | Han | East Guangdong | Shanwei | 1 |
| 40 | M | 32 | Han | East Guangdong | Shanwei | 1 |
| 41 | M | 50 | Han | East Guangdong | Shanwei | 1 |
| 42 | M | 25 | Han | East Guangdong | Shanwei | 1 |
| 43 | M | 48 | Han | East Guangdong | Jieyang | 1 |
| 44 | M | 44 | Han | East Guangdong | Jieyang | 1 |
| 45 | M | 37 | Han | East Guangdong | Jieyang | 3 |
| 46 | M | 39 | Han | East Guangdong | Jieyang | 1 |
| 47 | M | 48 | Han | West Guangdong | Yunfu | 3 |
| 48 | M | 44 | Han | West Guangdong | Maoming | 1 |
| 49 | F | 19 | Han | West Guangdong | Maoming | 5 |
| 50 | F | 39 | Han | West Guangdong | Zhanjiang | 4 |
| 51 | M | 47 | Han | West Guangdong | Zhanjiang | 1 |

*(Continued)*

**Table 2.** (Continued)

| Donor number | Sex | Age | Ethnicity | Birthplace | Residency | Number of blood donations |
|---|---|---|---|---|---|---|
| 52 | M | 30 | Han | West Guangdong | Zhanjiang | 10 |
| 53 | M | 35 | Han | West Guangdong | Zhanjiang | 1 |
| 54 | F | 43 | Han | West Guangdong | Zhanjiang | 3 |
| 55 | M | 32 | Han | West Guangdong | Zhanjiang | 1 |
| 56 | F | 49 | Han | West Guangdong | Zhanjiang | 6 |
| 57 | M | 53 | Han | West Guangdong | Zhanjiang | 5 |
| 58 | F | 48 | Han | North Guangdong | Meizhou | 5 |
| 59 | F | 19 | Han | East Guangdong | Shaoguan | 1 |

peptide of the gp46 protein was initially found and reported in China, it was presented in 5 (5/39, 12.8%) sequences of the Transcontinental subtype and 14 (14/16, 87.5%) sequences of Japanese subtype. The characteristic L55P mutation in the receptor binding domain of the gp46 protein was presented as 13 sequences (13/39, 33.3%) of the Transcontinental subtype, which has been reported in Fujian, China (Fig 4) [8].

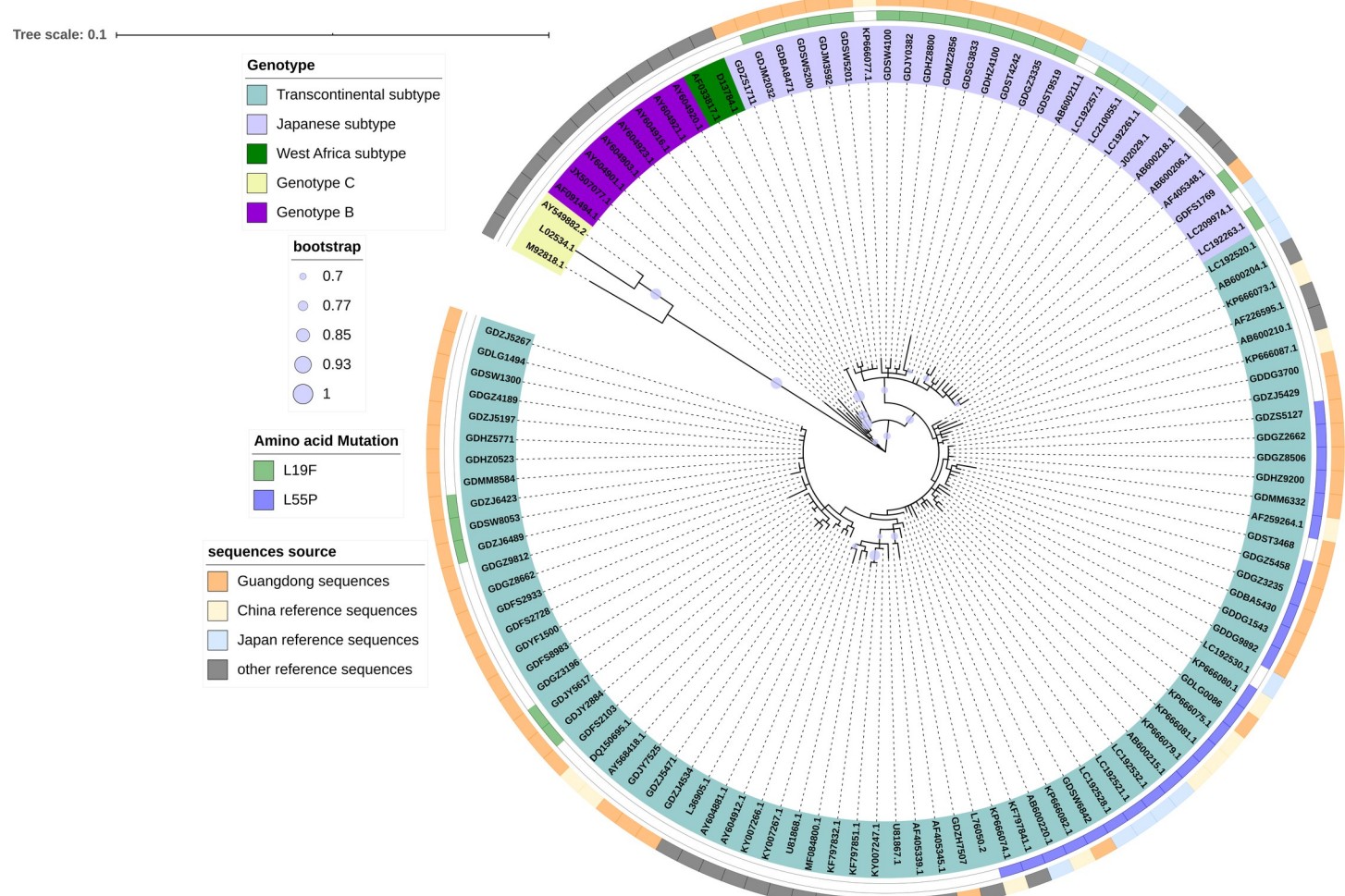

**Fig 2. Phylogenetic tree of HTLV-1 isolates based on gp46 sequences.** Support for the branching order was determined by 1000 bootstrap replicates; only values of 70% or more are shown.

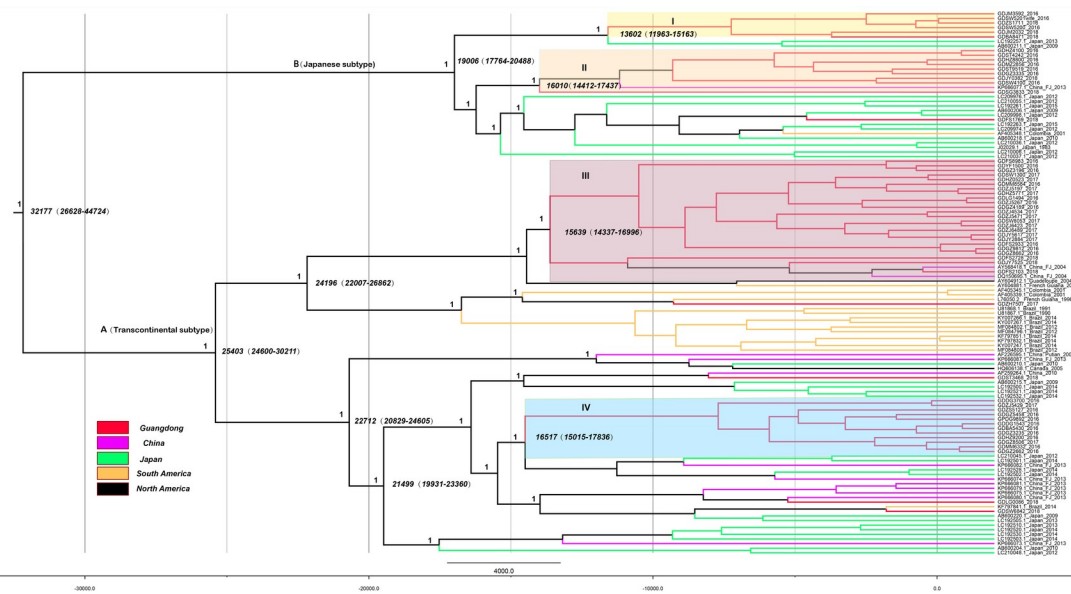

**Fig 3. MCC molecular clock phylogeny estimated based on the gp46 region.** The names of sequences from this study and reference consist of the isolate name, province, country and time of specimen collection. Branches were colored according to their sampling regions in different continents or countries/province. Time scale runs from 32177 years ago to 2018. Four clusters in the tree marked I, II, III and IV respectively showed the 4 different introduction events to Guangdong for HTLV-1. The posterior values of branches > 0.9 were shown only.

## Discussion

To the best of our knowledge, this was the most comprehensive epidemiology and evolutionary analysis study about HTLV infection among BDs in the Guangdong province of China. We screened 3,262,271 BDs in all cities of Guangdong province for three years and ultimately identified 59 confirmed cases of HTLV infection. The main findings of the current study include: 1) HTLV infection persists in many cities in Guangdong province, which is on the southeastern coast of China. 2) Only HTLV-1 infection was found. 3) The overall prevalence of HTLV in BDs was 1.81 per 100,000 in Guangdong, and 13.94 per 100,000 in Shanwei. Compared with the HTLV-1 prevalence in BDs with other countries, the prevalence in Guangdong province was lower than that in Japan (840 per 100,000) [13], Iran (2120 per 100,000) [14], Vietnam (6.3 per 100,000) [15], Brazil (24.7 per 100,000) [16], Peru (915 per 100,000) [17], the United States (21.9 per 100,000) [18], and other European countries excluding the Netherlands (0.85 per 100,000) [19].

According to limited reports [8,20,21], the Fujian, Guangdong and Zhejiang provinces in southeastern China are suggested to be HTLV prevalence regions with relatively high prevalence. Fujian province has the highest prevalence rate (16.9 per 100,000) in China's mainland [8]. According to the present study, the overall prevalence rate in Guangdong was lower than that in Fujian, however, Shanwei city which borders Fujian, presented the highest prevalence (13.94 per 100,000) in Guangdong, which was comparable with Fujian. The prevalence in Zhanjiang (5.31 per 100,000) was the second highest in Guangdong, although this city is located along the western edge of the province. Historically, many people migrated from Fujian to eastern Guangdong and Zhanjiang along the coastline in order to escape from war during the Tang and Song dynasties. Therefore, we speculate that the virus spread to eastern Guangdong and Zhanjiang after HTLV carriers migrated from Fujian, resulting in a high prevalence in Shanwei and Zhanjiang city. Further studies should be performed to investigate the

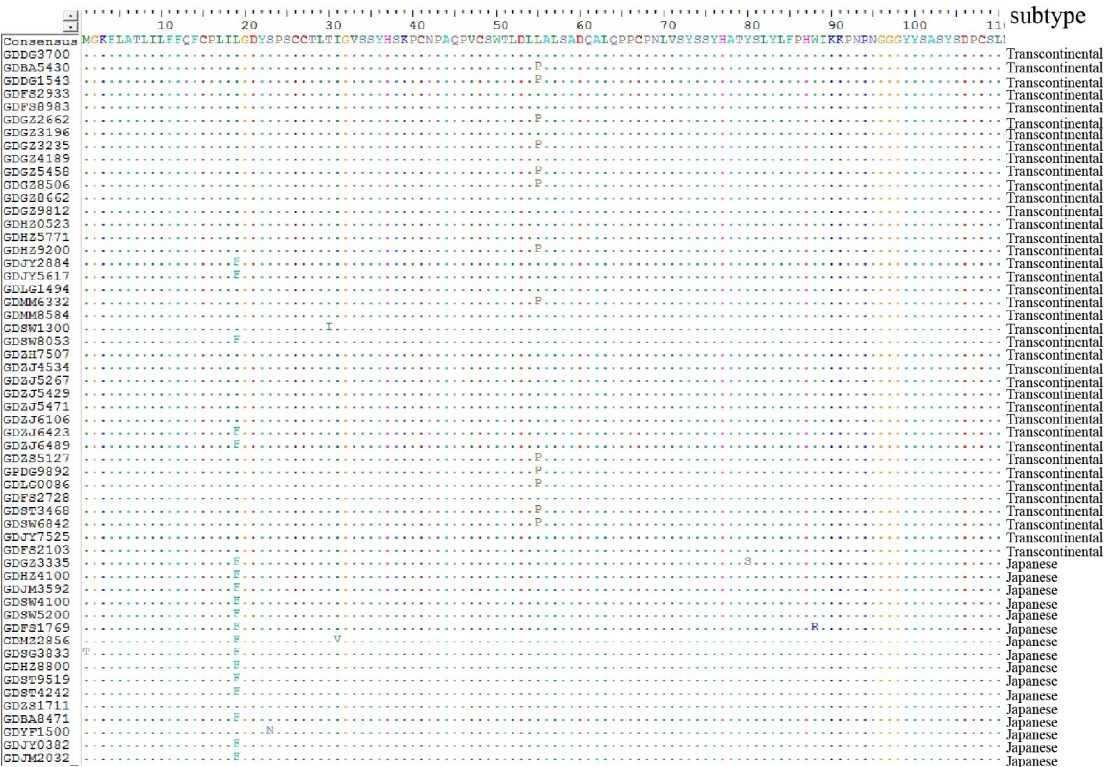

**Fig 4. Alignment of amino acids sequences of gp46 gene in strains isolated in this study.**

risk factors and transmission routes of HTLV-1 infection in these regions, with the aim of developing effective control strategies toward HTLV-1 infection.

The ML phylogenetic tree analyses in this study demonstrated that the majority of HTLV-1 isolates (39/55, 70.9%) in Guangdong belonged to the Transcontinental subtype. Furthermore, the remaining 16 isolates (16/55, 29.1%) belonged to the Japanese subtype. Previous studies have shown that the Japanese subtype is a predominant viral strain in the Japanese population, but it has also been found in Fujian and Taiwan [8, 22]. The presence of the Japanese subtype in the BDs of Guangdong indicated that this subtype may have originated from Japan.

Although the Transcontinental subtype distributed worldwide, previous studies[8] have reported that the HTLV-1 sequences of this subtype in mainland China, Japan, and Taiwan shared a unique gp46 L55P mutation with similar frequencies which was absent in sequences originating from other geographic regions. One-third of the sequences of the Transcontinental subtype first identified in Guangdong in the current study carried the gp46 L55P mutation, which was consistent with the previous report[8]. In our study, the first individual mutation of gp46 L19F in China was found, and this mutation was also mainly present in the reference sequences from Japan. Although the envelope glycoprotein gp46 is highly conserved among different HTLV-1 isolates, nucleotide substitutions in the region that codify these proteins may influence both the infectivity and replication of the virus. The gp46 gene has functional domains that have been associated with inhibiting of the formation of the syncytium, cell-cell transmission, and the production of antibodies [2,23,24]. The gp46 L55P mutation was only present in the Transcontinental subtype, which suggested that Transcontinental subtype produced unique protein variants for immune escape during transmission into East Asian populations. It also indicated that the virus in these regions may originate from a common

evolutionary ancestor. Nonetheless, the potential impact of this mutation on the expression and function of viral proteins deserves further investigation.

The HTLV-1 gene was highly conserved. We traced the Transcontinental and Japanese subtype dispersal based on the phylogenetic relationships of viral isolates from various ethnic groups. In this study, the gp46 gene MCC tree showed that the majority of sequences from Guangdong could be classified into clade I, II, III, and IV, all of which contained sequences only from China, which means that HTLV-1a migrated to China and formed four main virus strains and some sporadic epidemic strains. Cluster B showed that the Japanese subtype clusters in Guangdong originated from Japan. The earliest transmission of the virus cannot be verified, but from the root of our MCC tree, some sequence time nodes introduced into Guangdong were from earlier than 300 BC to 1400 AD. The historical records of China and Japan indicated that since the Han Dynasty, Japan has been in frequent contact with China, and this period coincided with the time node of the MCC tree. In addition, Kyushu, Shikoku, and Okinawa, areas with high HTLV prevalence in Japan, are adjacent to the southeastern coast of Guangdong. These factors have provided favorable conditions for the spread of HTLV. Previous studies [25,26] have shown that some HTLV-1 forms belong to a transcontinental subgroup that is related to Asian mongoloid HTLV-1. After excluding foreign sequences, we estimated that the tMRCAs of the Transcontinental subtype was earlier than Japanese subtype. This suggested that the Transcontinental subtype migrated into China before the Japanese subtype.

Our results (Table 1) showed that 54 of the 59 HTLV-1 infected individuals belonged to southeastern coastal cities and were characterized by HTLV mainly distributed along the southeastern coast. Among them, only 14 of the 33 infected people in the Pearl River Delta region were born locally, and the rest mainly came from Eastern Guangdong, Fujian, and other provinces. In particular, all infected people in Dongguan and Shenzhen were imported infection cases from other provinces. It can be hypothesized that HTLV-1 infections in the Pearl River Delta mainly came from other parts of China, especially Fujian and the adjacent eastern Guangdong regions. The Pearl River Delta is the most developed region in China's economy in terms of trade, science, education and health. Every year, hundreds of thousands of domestic and foreign travelers and businesspeople visit, and frequent population movements increase the risk of imported HTLV infection. The risk of infection may be reflected in the future from high-prevalence areas to low-prevalence areas, underdeveloped areas to developed areas, and net exporting areas to the net inflows of population.

In summary, the risk of transfusion transmission of HTLV in Guangdong province is lower than that of other viruses in Guangdong province, such as HCV (335 per 100,000) [27], HBV (130 per 100,000) [28] and HIV (66 per 100,000) [29], but it still calls for a significant concern for blood safety. It has been reported that leukocyte filtration has a protective effect on HTLV transfusion transmission [30]. Leukocyte filtration has been carried out about one-third of blood banks in China, while only five blood banks in Guangdong have carried out leukocyte filtration. The current implementation of leukocyte filtration cannot completely eliminate the risk of HTLV blood transfusion. Hence, HTLV screening in BDs may still be an important risk mitigation strategy. After years of experience and retrospective demonstrations, Scandinavian countries only screen first-time BDs [31]. In our study, 40 of the 59 infected donors in this study were first-time donors. This suggested that first-time BDs have a higher risk of transfusion-transmitted HTLV, which means after all repeat and regular BDs had finished screening, screening the first-time BDs alone can sufficiently minimize the risk of transfusion-transmitted HTLV. Considering cost-effectiveness, it may be reasonable to conduct screening in high-prevalence areas of Guangdong such as Shanwei and Zhanjiang. Low prevalence areas may be

considered for surveillance screening at intervals of five years to evaluate whether HTLV testing is required.

## Supporting information

**S1 Fig. Phylogenetic tree of HTLV-1 isolates based on LTR sequences.** Support for the branching order was determined by 1000 bootstrap replicates; only values of 70% or more are shown. Red circles indicated sequences from blood donors in this study.
(TIF)

**S2 Fig. MCC molecular clock phylogeny estimated based on the LTR region.** The names of sequences from this study and reference consist of the isolate name, province, country and time of specimen collection. Branches were colored according to their sampling regions in different continents or province. Time scale runs from 11744 years ago to 2018. Four clusters in the tree marked I, II, III and IV respectively showed the 4 different introduction events to Guangdong for HTLV-1. The posterior values of branches > 0.9 were shown only.
(TIF)

## Acknowledgments

We are thankful for all related staff from involved 24 blood establishments all over Guangdong.

## Author Contributions

**Data curation:** Qiao Liao, Zhengang Shan, Min Wang, Jieting Huang, Ru Xu, Tingting Li, Wenjing Wang.

**Investigation:** Qiao Liao, Zhengang Shan.

**Methodology:** Qiao Liao, Zhengang Shan.

**Project administration:** Xia Rong, Yongshui Fu.

**Writing – original draft:** Qiao Liao.

**Writing – review & editing:** Qiao Liao, Chengyao Li, Xia Rong, Yongshui Fu.

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
