## [Decision Letter · Decision Letter 0]

28 Jul 2020

Dear Dr. Fu,

Thank you very much for submitting your manuscript "Prevalence and Evolutionary Analyses of Human T-Cell Lymphotropic Virus in Guangdong Province, China: Transcontinental and Japanese Subtype Lineages Dominate the Endemic" for consideration at PLOS Neglected Tropical Diseases. As with all papers reviewed by the journal, your manuscript was reviewed by members of the editorial board and by several independent reviewers. In light of the reviews (below this email), we would like to invite the resubmission of a significantly-revised version that takes into account the reviewers' comments. 

We cannot make any decision about publication until we have seen the revised manuscript and your response to the reviewers' comments. Your revised manuscript is also likely to be sent to reviewers for further evaluation.

Sincerely,

Johan Van Weyenbergh

Associate Editor

Samuel Scarpino

Deputy Editor

Reviewer #1: The molecular clock has been used on gp46 with limited caution. It chould be completed with a study on other viral sequences (LTR) and the ESS for each node should be presented.

The legends of the phylogenetic trees that are presented lack information. Many more sequences available on genebank should be added.

Reviewer #1: While the epidemiological aspect of the study seems sound, the part concerning datation is very insufficient.

Reviewer #1: There are different types of conclusions that are presented :

- the region has some HTLV, however can a region with 1/100.000 be considered as endemic?

- the datations are conclusions on movemeent of population considers that the transcontinental virus had been introduced only once in the region, which is not evident.

Reviewer #1: Authors have determined the seroprevalence of HTLV among blood donors in the Guangdong region, China. The conclusion is that the prevalence is low. Phylogenetic analyses demonstrate that circulating strains belong either to TC or Japanese subclades. While the study is of interest, it is unclear whether the region can be considered as endemic (as the prevalence of very low) and datation methods and interpretations are very hazardous.

Reviewer #2: Apart from some attention to the phylogenetic aspects of this work, the manuscript needs some language editing to correct for the many linguistic errors. For example, abstract, last sentence "reduce the risk of its infection" does not correctly word what is meant. There are many such issues throughout the manuscript, which at times make it unclear what exactly is meant by the authors.

Reviewer #2: I am not an expert on HTLV epidemiology, which is why I limit my comments to the phylogenetic aspects of this paper.

Abstract

=========

- "using Bayesian Markov chain Monte Carlo (MCMC) inference method." => to my knowledge, MCMC is always "Bayesian". Why not rephrase to "in a Bayesian phylogenetic framework" or something alike?

- "The phylogenetic analyses showed that HTLV-1A is spread into China (18202 years ago)and became endemic." => it is well possible that undersampling underlies the lack of apparent spread from China to Japan. The authors should rephrase references to this in a more nuanced manner. 

- 21 cities are mentioned in the abstract while HTLV positive samples were detected in only 16 cities (line 144)?

Methods/results/discussion

- I am mostly concerned that with this combination of evolutionary rate and time scale, there can be a time-dependent effect on the evolutionary rate. As this can substantially impact the divergence datings, the authors should test for this by comparing the fit of a relaxed clock model (the model that was used by the authors) with that of a model that incorporates a time-dependent effect. For an example of this I refer to Membrebe et al, Mol Biol Evol 2019. 

- how was the mixing of the chains evaluated?

- did the authors use a skyline model, or a constant popsize model? There seems to be some ambiguity on this between he methods and results section. 

- "18,202 years ago (95% CI: 22,340-89,339)" => how is it possible that the mean or median estimate does not lie within the 95% CI? This looks like an error (or typo?).

- "Phylogeographic tree analysis of HTLV-1A genotype revealed different

180 relatedness of the isolates from China and those from other countries" => what exactly is meant by this?

- an objective criterium should be used to classify strains as 'group A or B'. Which one was used?

- "The major proportion of isolates from Guangdong were classified into clade I, II and III, which only contained isolates from China. Consequently, it is a China-specific clade. " => for such statements it is always good to keep in mind that undersampling can lead to a false image of population structure, or erroneous apparent migration links (see for example https://pubmed.ncbi.nlm.nih.gov/30118874/). I hence encourage the authors to make less strong statements when it comes to this. 

Also, as it is not possbile to analyse a complete sample, the number of apparent introductions into China always will be a minimal estimate. As the authors seem to have overlooked that individual lineages can also represent introductions (fig 3), I suggest that they formally estimate the number of introductions into China using Markov jumps (can be set up in BAEUTi), of that they instead take a parsimonious approach to count the number of introduction events.

- it would increase the ease of interpretation of results if the authors could also indicate the genotype A subtypes (transcontinental etc) and the type of amino acids at gp46 positions 19 and 55 in the MCC or ML tree. A nice visualisation tool to this end is iTol (https://itol.embl.de/). The phylogenies should also be annotated with branch support. 

-"Due to the genomic stability of HTLV-1," => what do the authors mean?

- "The communications between China and Japan are frequent and expatriates in China from Japanese enterprises are mostly in Guangdong." => the authors should discuss whether or not for HTLV the evolutionary and epidemiological events occur on the same time scale. For example, one could argue that given the 'deep' timing of the relevant branch(es) in the phylogeny, the explanation for the migration links should be historical rather than contemporaneous.
---

## [Decision Letter · Decision Letter 1]

8 Nov 2020

Dear Dr. Fu,

Thank you very much for submitting your manuscript "Prevalence and Evolutionary Analyses of Human T-Cell Lymphotropic Virus in Guangdong Province, China: Transcontinental and Japanese Subtype Lineages Dominate the  Prevalence" for consideration at PLOS Neglected Tropical Diseases. As with all papers reviewed by the journal, your manuscript was reviewed by members of the editorial board and by several independent reviewers. The reviewers appreciated the attention to an important topic. Based on the reviews, we are likely to accept this manuscript for publication, providing that you modify the manuscript according to the review recommendations. 

Sincerely,

Johan Van Weyenbergh

Associate Editor

Samuel Scarpino

Deputy Editor

Reviewer's Responses to Questions

**Key Review Criteria Required for Acceptance?**

**Methods**

-Are the objectives of the study clearly articulated with a clear testable hypothesis stated?

-Is the study design appropriate to address the stated objectives?

-Is the population clearly described and appropriate for the hypothesis being tested?

-Is the sample size sufficient to ensure adequate power to address the hypothesis being tested?

-Were correct statistical analysis used to support conclusions?

-Are there concerns about ethical or regulatory requirements being met?

Reviewer #1: When considering molecular clock on an encoding region (i.e. Env), authors should perform the analysis on the 3rd codon (there is indeed a different evolution rate depending on codons, the 3rd seem to be more reliable).

Considering PCR, authors should present the technique (nb of cycles, primer sequence).

**Results**

-Does the analysis presented match the analysis plan?

-Are the results clearly and completely presented?

-Are the figures (Tables, Images) of sufficient quality for clarity?

Reviewer #1: (No Response)

**Conclusions**

-Are the conclusions supported by the data presented?

-Are the limitations of analysis clearly described?

-Do the authors discuss how these data can be helpful to advance our understanding of the topic under study?

-Is public health relevance addressed?

Reviewer #1: It is well known that HTLV infection occurs on transfusion, but only is leucoreduction does not occur. Do blood banks in China perform leucoreduction? Please discuss

**Editorial and Data Presentation Modifications?**

Reviewer #1: AUthors should note the subtypes : HTLV-1a, -1b, 1c and not HTLV-1A, 1-B, 1-C. (abstract, and lines 171:172)

**Summary and General Comments**

Reviewer #1: The paper is interesting. For molecular clock, authors should add a study considering the 3rd codon of the coding sequence. Otherwise the manuscript is sound.

Please correct paraparesis (line 62), and prefer 'brest feeding' over 'breast milk' (line 65).

Line 72 is not clear: it is not because 'incubation period is long' that transfusion can occur. Please correct/
---

## [Editor Report · Decision Letter 2]

18 Dec 2020

Dear Dr. Fu,

We are pleased to inform you that your manuscript 'Prevalence and Evolutionary Analyses of Human T-Cell Lymphotropic Virus in Guangdong Province, China: Transcontinental and Japanese Subtype Lineages Dominate the  Prevalence' has been provisionally accepted for publication in PLOS Neglected Tropical Diseases.

Best regards,

Johan Van Weyenbergh

Associate Editor

Samuel Scarpino

Deputy Editor

---

## [Editor Report · Acceptance letter]

29 Jan 2021

Dear Dr. Fu,

We are delighted to inform you that your manuscript, " Prevalence and Evolutionary Analyses of Human T-Cell Lymphotropic Virus in Guangdong Province, China: Transcontinental and Japanese Subtype Lineages Dominate the  Prevalence," has been formally accepted for publication in PLOS Neglected Tropical Diseases.

Best regards,

Shaden Kamhawi

co-Editor-in-Chief

Paul Brindley

co-Editor-in-Chief
